# Small Extracellular Vesicles: Key Forces Mediating the Development and Metastasis of Colorectal Cancer

**DOI:** 10.3390/cells11111780

**Published:** 2022-05-29

**Authors:** Wenjie Zhang, Xiaoxue Hu, Zhengting Jiang

**Affiliations:** Clinical Medical College, Yangzhou University, Yangzhou 225000, China; 192201607@stu.yzu.edu.cn (W.Z.); 182201635@stu.yzu.edu.cn (X.H.)

**Keywords:** colorectal cancer, sEVs, TME, development, diagnosis, treatment

## Abstract

Colorectal cancer (CRC) is the third most common cancer worldwide, and its incidence and mortality rates have been increasing annually in recent years. A variety of different small extracellular vesicles (sEVs) are important mediators of intercellular communication and have an important role in tumor metastasis and progression. The development and metastasis of CRC are closely linked to tumor-cell-derived sEVs, non-tumor-cell-derived sEVs, and intestinal-microbiota-derived sEVs. Numerous studies have shown that the tumor microenvironment (TME) is a key component in the regulation of CRC proliferation, development, and metastasis. These sEVs can create a TME conducive to CRC growth and metastasis by forming an immunosuppressive microenvironment, remodeling the extracellular matrix, and promoting tumor cell metabolism. Therefore, in this paper, we review the role of different types of sEVs in colorectal cancer development and metastasis. Furthermore, based on the properties of sEVs, we further discuss the use of sEVs as early biomarkers for colorectal cancer diagnosis and the potential for their use in the treatment of CRC.

## 1. Background

Colorectal cancer is the third leading cause of cancer deaths and a major burden for health systems and patients globally [1]. For instance, the incidence and mortality rates of colorectal cancer in 2018 were 6.1% and 9.2%, respectively [2]. The transformation of normal epithelial mucosa into a hyperproliferative epithelium indicates the onset of colorectal cancer. These hyperproliferating intestinal epithelial cells (IECs) form adenomas after losing their normal tissue and structure, invading the lower mucosa, and they eventually spread to the colorectal system, becoming cancerous [3]. The TME is a complex, dynamic network of cells consisting of tumor cells, various types of immune cells (T cells, B cells, tumor macrophages, and NK cells), fibroblasts, adipocytes, extracellular matrix, microbiota, and a variety of cytokines and metabolites [4,5]. The TME promotes the development and proliferation of tumor cells by recruiting immunosuppressive cells, remodeling the extracellular matrix, and promoting angiogenesis [6]. Furthermore, abnormalities such as hypoxia, metabolic disturbances, and oxidative stress are evident in the TME. These abnormalities further suppress normal immune function and promote stromal fibrosis, which ultimately leads to tumor cell proliferation and metastasis. In CRC, there is little and phenotypically dysregulated infiltration of normal T and NK cells, resulting in the suppression of normal anti-tumor functions [7]. At the same time, myeloid suppressor cells and tumor-associated neutrophils (TAN), which are recruited in large numbers in the TME, suppress normal immune function by activating metalloproteinases, creating an immunosuppressive microenvironment [8]. In addition, pro-angiogenic factors secreted by CRC tumor cells promote the generation of dense vascular tissue that, together with the dense fibrotic network, leads to hypoxia and metabolic disturbances in the TME [9]. These factors interact and ultimately promote CRC proliferation and metastasis.

sEVs have a diameter of 30–150 nm and are found in all cells, including epithelial cells, endothelial cells, neuronal cells, fibroblasts, immune cells, and cancer cells [10]. They are surrounded by a lipid bilayer and contain typical biomolecules, including DNA, RNA, glycans, lipids, proteins, and metabolites [11]. sEVs can transfer the proteins and nucleic acids into the microenvironment where tumors grow, thus promoting tumor cell growth and metastasis [12]. sEVs promote tumorigenesis and metastasis by mediating intercellular communication in three ways: first, sEVs bind to target cell membrane proteins through their membrane proteins, thereby opening up signaling pathways in the target cell. Second, sEVs are cleaved by proteases; thus, the exosomal membrane proteins bind to receptors on the cell membrane to activate intracellular signaling pathways. Third, sEVs carry proteins, mRNAs, and microRNAs that can fuse with target cell membranes through the exosome membrane and exert their corresponding biological effects [13,14]. Tumor-cell-derived sEVs promote colorectal cancer (CRC) development and metastasis by forming an immunosuppressive microenvironment, remodeling the extracellular matrix, and promoting cellular metabolism [15]. M2 macrophage-derived sEVs and cancer-associated fibroblast (CAF)-derived sEVs also play a key role by remodeling the extracellular matrix and helping CRC tumor cells to accomplish immune evasion. The intestinal microbiota plays a crucial role in colorectal cancer development, progression, and metastasis. The intestinal microbiota enters the body’s circulatory system through its derived sEVs and becomes implanted in other organs, providing a suitable pre-metastatic microenvironment for colorectal cancer cell metastasis. Intestinal-microbiota-derived exosomes promote CRC by breaking the outer intestinal mucosal barrier, producing chronic inflammation, and mediating immunosuppression.

This study highlights the effects of tumor-derived sEVs, M2 macrophage-derived sEVs, CAF-derived sEVs, and intestinal-microbiota-derived sEVs on CRC progression and metastasis. This study also focuses on the use of sEVs in the diagnosis and treatment of colorectal cancer.

## 2. Tumor-Derived sEVs Reshape the CRC Tumor Microenvironment

The tumor microenvironment (TME) plays a crucial role in cancer progression and metastasis [16]. Immune cells, fibroblasts, and endothelial cells are the main cells in the TME of colorectal cancer [17]. sEVs, as key mediators of intercellular communication, mediate communication between tumor cells and the surrounding microenvironment as well as cells from distant organs [18]. Here, the role of tumor-derived sEVs in the surrounding microenvironment of CRC consists mainly of forming an immunosuppressive environment, remodeling the extracellular matrix, and influencing cellular metabolism, as has been discussed. The mechanisms by which sEVs are involved in these processes have also been explained.

### 2.1. sEVs Mediate the Formation of Immunosuppressive Environments in Distant Organs

Tumor-cell-derived sEVs promote CRC development and metastasis by mediating the formation of an immunosuppressive microenvironment in distant organs. sEV-mediated immunosuppression mainly involves the inhibition of immune cell proliferation and the recruitment of immunosuppressive cells [19].

(1) The inhibition of the proliferation and function of normal immune cells. Immune cells consist of T and B lymphocytes, natural killer (NK) cells, and tumor-associated macrophages (TAM) [20]. The NK cells recognize, respond to, and directly defend against invading tumor cells; B cells combined with T helper cells can inhibit tumor progression by producing tumor-specific antibodies. For instance, CD8 cytotoxic T lymphocytes (CTL) can directly destroy tumor cells [21]. sEVs carrying PD-L1 enter the lymphatic system from the bloodstream, preventing T cell activity. As a result, immune cells cannot recognize and remove tumor cells [22,23]. Tumor sEVs can also interfere with monocyte differentiation and inhibit NK2D expression in normal NK cells [19]. Mutations in exosomal KRAS can lead to increased IL-8 production, neutrophil recruitment, and the formation of neutrophil extracellular traps (NETs), causing CRC development [24]. Tumor-derived sEVs can promote CRC development by carrying lncRNAs that promote the differentiation of CD+4 T cells to Th17 cells, which destroy associated immune cells [25].

(2) The recruitment of immunosuppressive cells to co-create an immunosuppressive microenvironment. sEVs can recruit regulatory T cells (Tregs) and bone-marrow-derived suppressor cells (MDSCs) to distant secondary sites to suppress anti-tumor immunity and promote tumor progression [21,26]. Tregs can suppress cytotoxic T cell responses and maintain immune tolerance by producing the immunosuppressive cytokines TGF-β and interleukin-10(IL-10) [27]. Similarly, MDSCs can inhibit T cell activation. Moreover, exosome-carried HSP72 can induce the STAT3-dependent immunosuppressive function of MDSCs, thus achieving immune tolerance [28]. Wang et al. showed that exosome S100A9 can promote MDSC accumulation in tumor tissue and mediate G-MDSC chemotaxis during colorectal carcinogenesis. Meanwhile, MDSCs can enhance the stemness of colorectal cancer cells in a partially exosome-dependent manner [29]. Tumor-associated macrophages (TAM), the most abundant cells in the TME, are crucial for remodeling the microenvironment surrounding tumor cells, accelerating the formation of an immunosuppressive TME [30,31]. For instance, M1-type macrophages are involved in anti-tumor immunity, while M2 macrophages promote the immune evasion capabilities of CRC by generating anti-inflammatory signals [32], similar to what was shown in several studies [33,34]. For instance, Wang et al. demonstrated that several exosome-encapsulated miRNAs (miR-25-3p, miR-130b-3p, and miR-425-5p) can induce M2 polarization in macrophages by activating the PI3K/AKT signal pathway, thereby promoting CXCL12/CXCR4-induced colorectal cancer liver metastasis (CRLM) [33]. Zhao et al. found that the CRC-derived exosome miR-934 can induce M2 macrophage polarization by downregulating PTEN expression and activating the PI3K/AKT signaling pathway. They also showed that polarized M2 macrophages could promote CRLM through a CXCL13/CXCR5/NFκB/p65/miR-934 positive feedback loop [34]. These results indicate that sEVs promote an immunosuppressive environment by suppressing immune cells and recruiting immunosuppressive cells. Moreover, sEVs combined with TAMs can promote the development of colorectal cancer. Here, we summarize the role of CRC-tumor-cell-derived sEVs in shaping the immunosuppressive microenvironment (Figure 1).

### 2.2. sEVs Are Involved in Remodeling the Extracellular Matrix of CRC

The extracellular matrix (ECM) is a non-cellular component contained in all tissues. Its immune and stromal components play a key role in the metastasis and progression of colorectal cancer [35,36]. Proteins and other RNAs on the surface of sEVs induce inflammation, fibrosis, and damage to the ECM, thus metastasizing tumor cells to distant organs. The CAFs are the major stromal cells in CRC tissues. They are involved in tumor progression [37], tumor metastasis, and drug resistance through paracrine factors [38]. sEVs promote tumor progression and metastasis through dynamic crosstalk between CAFs and cancer cells, thus inducing the effectiveness of anti-tumor therapy [39]. Importantly, CAFs can deposit ECM proteins and secrete growth factors, as well as contract and remodel the ECM through growth factors, chemokines, miRNAs, and other components carried by sEVs, ultimately leading to fibrosis development [40,41]. Tumor-cell-derived sEVs play a crucial role in remodeling the ECM. For instance, Mercedes Herrera et al. demonstrated, using bioinformatic analysis of sEVs loaded with ncRNA regulatory elements, that CAF-derived sEVs can mediate specific crosstalk between CAFs, colorectal cancer cells, and other stromal cells [42]. Jing Ren et al. also found that CAF-derived sEVs can recruit H19, which is highly expressed in CAFs, to activate the β-linked protein pathway by acting as a competitive endogenous RNA sponge for miR-141 in CRC. miR-141 can significantly inhibit the stemness of CRC cells. Therefore, H19 can enhance the stemness of colorectal cancer cells by blocking miR-141 [43]. Yang et al. recently showed that CAFs can promote the growth and metastasis of CRC by inducing the exosome circEIF3K [44]. Furthermore, the exosome miR-590-3p from CAFs can enhance the resistance of CRC cells to radiation therapy through the CLCA4/PI3K/Akt axis, thus providing immunosuppression against tumor therapy [39]. The lncRNA LINC00659 can be transferred from CAFs to CRC cells via sEVs, thus promoting CRC development through ANXA2 upregulation [45]. Similarly, the CAFs-derived sEV microRNA-24-3p can enhance the resistance of colorectal cancer cells to MTX by downregulating the CDX2/HEPH axis [46]. Notably, specific exosomal integrins can interact with the extracellular matrix of deposited laminin and fibronectin, thus leading to increased adhesion to the extracellular matrix and the colonization of circulating tumor cells [47]. sEVs induce CAFs to remodel the ECM and form an immune-cell-permeable barrier by building a link between CAFs and cancer cells, thus providing a suitable environment for the survival of CRC tumor cells (Figure 2).

### 2.3. Tumor-Derived sEVs Promote the Metabolism of Promoted Tumor Cells

The increased bioenergetic and biosynthetic demands of tumor cells depend on various metabolic pathways autonomously altering their fluxes [48]. sEVs can access metabolic information from adjacent or distant cells to facilitate the release of bioactive molecules such as vascular endothelial growth factor (VEGF), lipids, and lactate [49,50]. First, angiogenesis is a key factor promoting the proliferation and development of CRC tumor cells. Although normal angiogenesis is essential for development and tissue growth, tumor-induced angiogenesis provides the oxygen and nutrients necessary for the growth and spread of cancer. Tumor-induced angiogenesis also removes waste products [51]. Angiogenesis in colorectal cancer is characterized by a high density of microvessels that accumulate in a well-defined area of the primary tumor close to the intestinal lumen [52,53]. sEVs can regulate endothelial cell properties by transporting many pro-angiogenic biomolecules, such as vascular endothelial growth factor (VEGF), matrix metalloproteinases (MMP), and microRNAs, thus promoting angiogenesis during cancer progression, particularly under hypoxic conditions [48,54]. The miR-25-3p secreted by CRC can be transferred to human umbilical vein endothelial cells (HUVECs) to regulate vascular permeability and angiogenesis by silencing KLF2 and KLF4. This process requires exosomal involvement [55]. Tumor-cell-derived exosomes carrying MiR-21-5p can inhibit the interaction between Krev and capture protein 1 (KRIT1) in HUVECs and activate the β-linked protein signaling pathway by increasing the downstream targets VEGFa and CCND1, thereby promoting angiogenesis and vascular permeability in CRC [56]. CRC-cell-derived sEVs can promote the proliferation, invasion, and tube formation capacity of HMEC-1 cells by abnormally increasing miR-183-5p levels, thus inducing angiogenesis through FOXO1 downregulation [57]. Huang et al. found that exosome-mediated Wnt/β-linked protein signaling can tightly link endothelial cells and CRC cells under hypoxia, thereby promoting angiogenesis [58]. Several studies have also found that sEVs are closely associated with other bioactive factors and, thus, can promote cellular metabolism around tumor tissue by interacting with these bioactive factors. Wei Wang et al. demonstrated that sEVs carrying miRNAs can prevent ribonucleases from attacking and thus crosstalking with lipids and their modifying proteins, enzymes, and miRNAs [49]. Glycolysis enables rapid fluctuations in energy demand, thereby promoting tumor cell proliferation and metastasis. Exosome-transported cyclic RNA-122 (ciRS-122) derived from oxaliplatin-resistant CRC can promote glycolysis and drug resistance by upregulating pyruvate kinase M2 (PKM2) expression in sensitive cells [59]. Lactate creates an acidic environment, essential for tumor cell metastasis, angiogenesis, and resistance to treatment. Ma et al. found that U251-cell-derived sEVs can induce lactate production by upregulating Glut-1, HK-2, and PKM-2 levels [60]. The hypoxic TME induces a glycolytic phenotype in many cancer cells and characterizes colorectal cancer development. Recent studies have found that miR-410-3p-rich sEVs can crosstalk with the hypoxic microenvironment of CRC with aerobic cells to enhance tumor progression [61]. Similarly, hypoxia can also upregulate the expression of CRC-derived exosomal miR-361-3p, thereby promoting CRC cell proliferation and inhibiting apoptosis [62]. These results indicate that tumor-cell-derived sEVs promote the release of bioactive factors (VEGF, lipids, and lactate) and the formation of a hypoxic microenvironment that accelerates the cellular metabolism around tumor tissue and the development and metastasis of CRC (Figure 3).

## 3. Other Cell-Derived sEVs Promote the Development and Metastasis of CRC

Besides tumor-cell-derived sEVs, M2 macrophage-derived sEVs and fibroblast-derived sEVs also play key roles in the development and metastasis of colorectal cancer. (1) M2 macrophage-derived sEVs can promote the development and metastasis of CRC cells in various ways, including remodeling the ECM and inducing the expression of immunophenotypic antigens in CRC cells. Cathepsin B, D, K, L, and S, as well as MMP-8, MMP-12, MMP-13, and integrin β1, α4 are highly expressed on M2 macrophage-derived sEVs. These bioinformatic molecules play crucial roles in remodeling the TME, indirectly suggesting that M2 macrophage-derived sEVs play a key role in CRC remodeling [63]. M2 macrophage-derived sEVs can also promote CRC invasion by inducing the epithelial-to-mesenchymal transition (EMT) of CRC cells. M2 macrophage-derived sEVs with a high expression of glypican-1 can induce EMT in patients with CRC by suppressing E-cadherin expression and promoting the expression of vimentin and snail [64]. M2 macrophage-derived sEVs can promote EMT formation by activating the Wnt/β-catenin axis. M2 macrophage-derived sEVs carry CD14, CD68, MAC387, CD163, and DAP12 phenotypic antigens that can be detected in CRC. These immunophenotypic antigens promote the complete immune evasion of tumor cells [65]. Moreover, M2 macrophage-derived sEVs carry miRNAs (miR-21-5p and miR-155-5p) that promote CRC progression and invasion by suppressing BRG1 expression [66]. (2) CAF-derived sEVs also promote CRC proliferation and invasion. CAF-derived sEVs can promote drug resistance and immune evasion in CRC by promoting the formation of fibrous networks in the ECM through growth factors and proteolytic enzymes [67]. CAF-derived sEVs inhibit mitochondrial apoptosis by promoting EMT in CRC cells and promoting the expression of miR-92a-3p in CRC cells, thereby activating the Wnt/β-linked protein pathway and promoting CRC metastasis [68]. Some cells (mesenchymal stem cells) derived from sEVs can promote the function of immune cells, reduce immunogenicity, and inhibit the development and metastasis of CRC. However, these should be further explored in future.

## 4. Intestinal Microbial-Derived sEVs Promote CRC Development and Metastasis

Many gastrointestinal-associated malignancies (gastric, pancreatic, and colorectal cancers) occur and progress without microbial ecological dysregulation [69]. Dysbiosis of the intestinal microbial ecology may trigger the release of sEVs. sEVs derived from intestinal microbiota are divided into outer membrane vesicles (OMVs) from Gram-negative bacteria and membrane vesicles (MVs) from Gram-positive bacteria, parasites, fungi, and mycobacteria, which have similar sizes, structures, and biological functions to sEVs from mammalian cells [70,71,72].

The sEVs derived from intestinal microbiota assume bacteria secretion and transport systems that can be linked through their cargo (DNA and RNA, proteins, lipids, and other bioactive molecules) to eukaryotic cells; they establish close links [73]. This study shows that intestinal-microbial-derived sEVs are strongly related to CRC development and metastasis. The highly permeable intestinal mucosal barrier is connected by tight junctions, avoiding direct contact between the intestinal microbiota and immune cells [74,75]. Intestinal-microbial-derived sEVs can achieve the invasion of parental pathogens into the intestinal epithelium and subsequently trigger inflammation by altering the composition of the tight junctions [76]. For instance, the vesicles of *Campylobacter jejuni* can enhance the invasion ability of *Chlamydomonas jejuni* by disassembling linker proteins, including *E-calmodulin* and *occludin* [77]. The development of intestinal inflammation can significantly promote CRC development. Intestinal-microbial-derived sEVs may initiate pro-inflammatory signaling and participate in the formation of pre-metastatic ecological niches for future tumor colonization by interacting with distant organ receptor cells via systemic circulation. Hickey et al. demonstrated that certain pro-inflammatory cytokines, such as TNF-α and IL-1β, can cause an inflammatory response through the induction by sEVs of the symbiotic bacterium θotaomicron, thereby promoting colitis development in genetically susceptible mice [78]. Cañas et al. also reported that sEVs of the symbiotic intestinal microbiota bacterium ECOR12 can modulate the intestinal immune response by enhancing the secretion of the pro-inflammatory cytokines IL-6 and IL-8 in Caco-2 cells [79]. Additionally, Patten et al. showed that sEVs derived from the symbiotic bacterium E. coli C25 can drive inflammatory responses in intestinal epithelial cells in vitro by secreting IL-8 [80]. Pathogens can suppress the host immune system through sEVs derived from intestinal microbiota to exacerbate infection or trigger an overextended immune response, leading to sepsis and cancer [81,82,83]. For example, sEVs released by *Lactobacillus* and *Bifidobacterium* spp. can activate the TLR 2/6 response [84]. In conclusion, sEVs from the intestinal microbiota play a key role in the development and metastasis of CRC by participating in the development of local inflammatory responses in the intestine and the formation of an immunosuppressive microenvironment at local tissue sites.

## 5. Application of sEVs in the Diagnosis and Treatment of CRC

### 5.1. sEVs as Early Markers for Colorectal Cancer Diagnosis

sEVs contain unique molecules, including DNA, RNA, microRNA, lncRNA, proteins, and lipids. As a result, they can be used as biomarkers for cancer diagnosis and to show specific gene expression profiles [85]. Biomarkers may be molecules secreted by tumors or specific responses of the body due to the presence of cancer. Plasma and other systems are widespread in the body, making sEVs mobile; thus, they can play a crucial role in identifying and predicting biomarkers of human cancers. For example, high levels of expression of highly enriched miRNA in colorectal cancer sEVs, such as plasma exosomal miR-125a-3p and miR-6803-5p, can increase diagnostic power and act as a biomarker for analysis in the initial stages of CRC [86,87]. Plasma sEV miR-92b is barely expressed in patients with CRC and highly expressed in healthy individuals and can be used as a biomarker for the diagnosis of CRC [88]. Plasma sEV miR-27a and miR-130a can be used as non-invasive biomarkers for the early detection and prognosis prediction of CRC [89,90]. A recent raw letter study conducted by screening key pathways and GO terms regarding up- and down-regulated transcripts identified the following diagnostic CRC biomarkers: lncRNA BLACAT1 and four down-regulated lncRNAs (LOC344887, LINC00675, DPP10-AS, and HAGLR) [91]. Another study showed that three exosomal lncRNA in plasma (LNCV6_116109, LNCV6_98390, and LNCV6_84003) are expressed completely differently in patients with CRC and healthy individuals and can be used as diagnostic biomarkers [92]. Liu et al. showed that exosomal lncRNA CRNDE-h expression is higher in the serum of patients with colonic adenomas than in normal subjects and thus could be used as a biomarker for CRC diagnosis [93]. Some sEVs carry the highly expressed lncRNA CCAT2 that promotes CRC metastasis and progression, while the lncRNA GAS5, when highly expressed, inhibits CRC tumor cell proliferation and invasion. Both are expressed dramatically differently in patients with CRC and healthy individuals and can be used as diagnostic and prognostic biomarkers for CRC [94,95]. Furthermore, Paul et al. demonstrated that the surface density of hyaluronic acid (HA) is significantly higher in sEVs from colorectal cancer cells than in normal cells, using high-resolution atomic force microscopy (AFM) and spectroscopy (AFS) techniques [96]. This suggests that sEVs from colorectal cancer cells are different from colonic epithelial cells at the level of single vesicles and thus could be used as a potential biomarker for CRC diagnosis. Tian et al. also analyzed blood samples using HSFCM and found that the level of CD147 positivity in sEVs was significantly higher in colorectal cancer patients than in healthy controls [97]. As a platform for the analysis and quantification of surface proteins in individual EVs, HSFCM has greatly enhanced the understanding of exosome-mediated intercellular communication and the development of advanced diagnostic strategies. A study showed that 36 proteins are upregulated and involved in the regulation of the metastatic microenvironment, while 22 proteins are downregulated and involved in tumor cell production and development in the serum sEVs of patients with CRC [98]. Another study indicated that heat shock protein 60 (Hsp60) from plasma sEVs can be used for CRC diagnosis [99]. The sEV-derived protein GPC1 is abundantly expressed in patients with CRC with high sensitivity, while sEV CPNE3 is abundantly expressed in CRC tissues and less expressed in other cancers with relatively high specificity. Based on the biological properties of both, they can be used as biomarkers for the diagnosis of CRC [100,101]. Some sEVs carry the protein SPARC, and its high expression often suggests a poor prognosis for CRC [102]. Plasma-sEV-derived CEA has high sensitivity and specificity in CRC diagnosis and can be used as an early biomarker for the diagnosis of CRC [103]. Herein, we summarize the value of different sEVs in the diagnosis of colorectal cancer. (Table 1).

### 5.2. The Value of sEVs in CRC Treatment

sEVs can act as carriers for the transfer of drugs to target cells due to their small size (nanoscale dimension); thus, they could play a crucial role in CRC therapy. For example, sEVs containing azithromycin prevent the growth of CRC cells in specific cases by delivering drugs to the target organ [104]. sEV miRNA-375 can prevent the spread of tumor cells by blocking Bcl-2 in CRC, suggesting that engineered sEVs may be a possible therapeutic target for CRC by significantly reducing cancer progression [105]. Furthermore, miRNA-contained sEVs can be used to overcome drug resistance arising from cancer therapy. For instance, miR-214 enhances the radiosensitivity of CRC by inhibiting autophagy in CRC cells [106]. In summary, sEVs are crucial for CRC treatment.

Intestinal-microbiota-derived sEVs are useful in CRC treatment due to their inherent properties, such as size, antigenic stability, high immunogenicity, accurate host cell targeting, capacity for specific cargo delivery, and host immune response [107,108,109]. A recent study also demonstrated that intestinal-microbial-derived sEVs have potential in cancer immunotherapy. For instance, Kim et al. selectively and thermally targeted tumor tissue and induced a long-term anti-tumor immune response through the production of the cytokines CXCL10 and interferon-gamma by intravenously and systemically administering sEVs of Gram-negative bacteria from genetically modified *E. coli msbB*, thus completely eradicating the established tumor without significant adverse effects. Similar anti-tumor effects have been observed for sEVs from *Lactobacillus acidophilus* and *Staphylococcus aureus* [110]. sEVs derived from intestinal microbiota can also be used to prevent or treat inflammatory diseases due to their anti-inflammatory properties, thereby reducing the likelihood of CRC. For example, sEVs derived from *Bacteroides fragilis* promote the secretion of anti-inflammatory cytokines and inhibit the secretion of pro-inflammatory cytokines [111]. Intestinal-microbial-derived sEVs, genetically engineered to modify bacteria and subsequently purified and recombinant, are potential cancer vaccines. Intestinal-microbial-derived sEVs can trigger powerful and durable anti-tumor immune responses, even in combination with CTLA-4 and anti-PD1 immunotherapies based on surface decoration with multiple heterologous tumor antigens and immunostimulatory bacterial DNA (CpG motifs) [112]. sEVs can prevent and treat inflammatory, autoimmune, and metabolic diseases associated with CRC development by transporting miRNAs and regulating the intestinal microbiota [113,114,115]. miR-1226-5p and miR-515-5p can reshape the intestinal microbiota by promoting the growth of *Escherichia coli* and *Clostridium perfringens* nuclei, respectively, thereby avoiding microbiota dysbiosis and CRC development [116]. Despite sEVs’ potential to play a crucial role in CRC treatment, sEVs are associated with some problems, possibly related to targeting and purification methods. Therefore, relevant clinical trials should be refined and improved. This study summarizes the current clinical trials on the use of sEVs in the diagnosis and treatment of CRC, providing insights for future work (Table 2).

### 5.3. Clinical Implications of sEVs as Markers and Drug Carriers in CRC Screening, Diagnosis, Treatment, and Prognosis

sEVs have very obvious advantages as biomarkers for the early diagnosis, diagnostic staging, efficacy monitoring, and prognostic assessment of CRC. (1) sEVs have a very high clinical value in the early screening of CRC. Ren et al. found that sEV-derived miR-196b-5p was highly differentially expressed in healthy individuals and patients with CRC, where sEV-derived miR-196b-5p was highly expressed in patients with CRC, but hardly expressed in healthy individuals, and its accuracy in screening for CRC was 0.88 [117]. (2) sEVs showed advantages far beyond other biomarkers for the diagnosis of CRC. Donatella et al. found that sEVs were more sensitive than circulating tumor DNA (ctDNA) and circulating tumor cells (CTC) for the diagnosis of CRC, with higher sensitivity and specificity [118]. Furthermore, human plasma sEV-derived miRNAs had higher specificity than total plasma miRNAs in the early diagnosis of colon cancer. Li et al. found that plasma sEV miRNA (Let-7b-3p, miR-130-3P, miR-145-3P, and miR-139-3p) were much more accurate than total plasma miRNAs for the early diagnosis of patients with CRC. The area under the curve (AUC) for the four markers combined was 0.932 [119]. Another study showed that the sensitivity of sEV-derived miR-1246 and miR-23a in CRC diagnosis was 92% and 95.5%, respectively [120]. (3) sEVs have great potential as biomarkers for CRC prognostic analysis. Tan et al. found that reduced expression of serum sEV miRNA-199ashowed a positive correlation with poor CRC prognosis, and affected patients tended to have lower survival rates [121]. Another similar study showed that the sEV-carrying lncRNA MALAT1, at high levels of expression, showed a significant correlation with poor patient prognosis; in addition, its high expression was strongly associated with poor quality of life and resistance to chemotherapeutic agents (oxaliplatin) in patients with advanced CRC [122]. In patients with CRC, increased levels of sEV miR-429 expression were significantly and positively correlated with lower OS and poor prognosis [89]. (4) sEVs can be used as drug carriers and ribonucleic acid therapeutic agents for the treatment of CRC. sEVs, as a special targeted drug delivery system, can facilitate more drug flow to tumor cells, avoiding phagocytosis by macrophages and cleavage by some special enzymes. In treatments to prevent the development of CRC, the use of the targeted delivery of sEVs carrying adriamycin to the target organ can have significant preventive effects, and the incidence of CRC is greatly reduced [104]. When loaded with paclitaxel on milk-derived sEVs, modified by relevant techniques, and then infused back into humans, the tumor-suppressive ability of paclitaxel was significantly enhanced and produced fewer side effects [123]. sEVs delivering small molecule RNAs have shown great value in CRC therapy. sEVs rely on small molecule RNA therapies such as small interfering (siRNA) and miRNA therapies. sEVs are used as nano-delivery systems loaded with siRNAs for targeted delivery to target cells. The sEV-based delivery systems are far superior to other delivery systems in inhibiting tumor growth [124]. Loading miRNA-21 and chemotherapeutic drugs (5-fluorouracil (5-FU)) onto sEVs significantly enhanced the anti-tumor effect of 5-FU on CRC [125]. Through nanoengineering techniques, sEVs carrying miR-214 could significantly enhance the sensitivity of radiotherapy and inhibit the growth of CRC [106]. sEVs loaded with miR-25-3p inhibitors could significantly inhibit the vascular permeability of CRC tumor tissues and suppress tumor growth and metastasis in patients with CRC [55]. There are already a large number of clinical trials investigating the clinical role of sEVs in the early detection, diagnosis, efficacy monitoring, and assessment of the prognosis of CRC. For example, in a search through Clinicaltrials.gov, one finds NCT04523389, NCT04394572, NCT01294072, and other relevant clinical trials that focus on the clinical feasibility of applying sEVs for CRC diagnosis and prognosis and as drug carriers for treatment. Of course, sEVs currently have some uncertain risk factors, such as the question of whether they will continue to be efficacious and safe for long-term application. The purification and isolation of sEVs are also big challenges at present, and these limitations deserve to be explored in future research.

## 6. Conclusions

In this review, we summarized the role of sEVs as a key force promoting CRC onset, progression, and metastasis. Tumor-derived sEVs mediate intercellular communication and shape the microenvironment surrounding colorectal cancer in various ways, including shaping the immunosuppressive environment and remodeling the ECM and cellular metabolism. Intestinal-microbiota-derived sEVs drive CRC development by promoting inflammation and immunosuppression. Finally, we summarized the potential use of sEVs as biomarkers for the early diagnosis of colorectal cancer and the use of sEVs in combination with other therapeutic approaches to treating colorectal cancer based on TME-based research. Therefore, future sEV research may provide further insights into the mechanisms of CRC development and improve the diagnosis and treatment of CRC.

## Figures and Tables

**Figure 1 cells-11-01780-f001:**
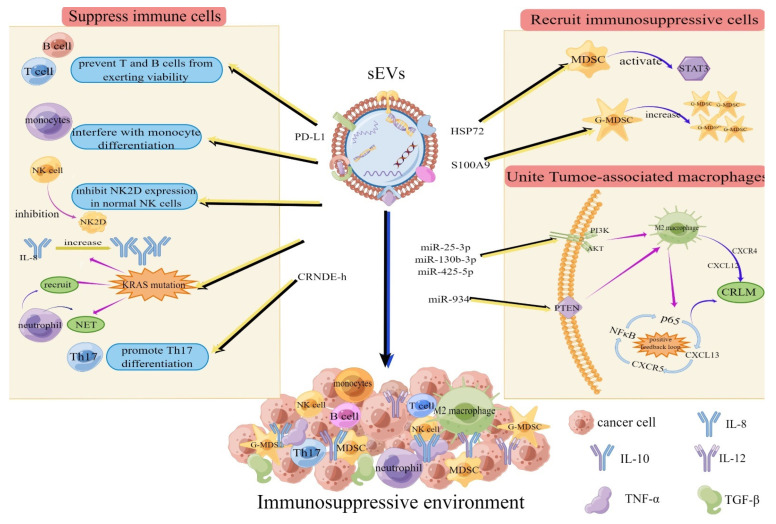
sEVs promote the formation of an immunosuppressive environment. sEVs form an immunosuppressive environment by inhibiting immune cells, recruiting immunosuppressive cells, and combining with TAMs. sEVs inhibit immune cells by PD-L1 causing immune cells to lose the ability to recognize and clear tumor cells, interfering with monocyte differentiation, and inhibiting NK2D expression in normal NK cells. KRAS mutation leads to increased IL-8 production, neutrophil recruitment, and the formation of neutrophil extracellular traps (NET). CONDE-h promotes Th17 cell differentiation. The methods by which sEVs recruit immunosuppressive cells include: HSP72 induces STAT3-dependent immunosuppressive functions of MDSC, and S100A9 promotes G-MDSC proliferation. The methods by which sEVs combine with TAMs are as follows: miR-25-3p, miR-130b-3p, and miR-425-5p induce the M2 polarization of macrophages by activating the PI3K/AK signaling pathway, thereby promoting the CXCL12/CXCR4-induced liver metastasis of colorectal cancer (CRLM). Mir-934 induces the polarization of M2 macrophages by down-regulating PTEN expression and activating the PI3K/AKT signaling pathway; thus, it forms a positive feedback loop of CXCL13/CXCR5/NFκB/P65/Mir-934 to promote CRLM. Explanation: MDSC: Myeloid-derived suppressor cells, PD-L1: Programmed death-ligand 1.

**Figure 2 cells-11-01780-f002:**
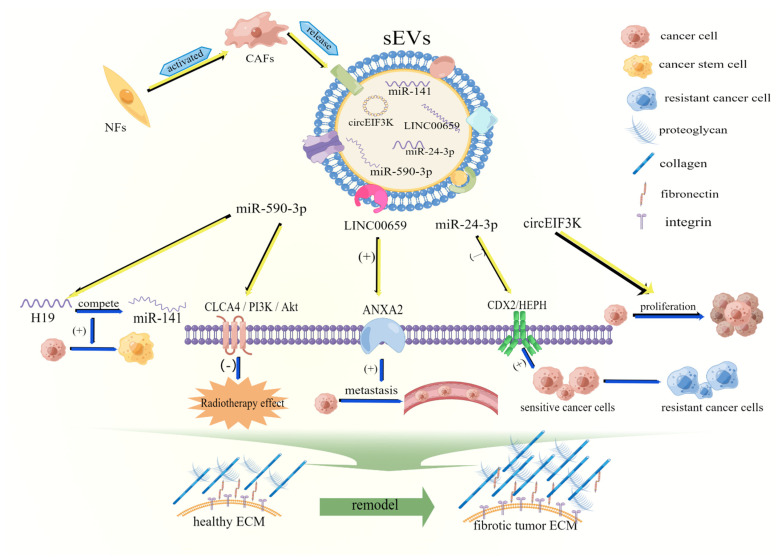
sEV mechanisms of remodeling the extracellular matrix. After NF activation into CAFs, CAF−derived sEVs can regulate tumor cell proliferation and enhance the viability and drug resistance of colorectal cancer cells. Additionally, miRNA carried by sEVs can enhance resistance to radiotherapy via the CLCA4/PI3K/Akt axis. In addition, lncRNA LINC00659 can promote the TME development of CRC by transferring from CAFs to CRC cells through sEVs and upregulating ANXA2 expression. sEVs can reshape the extracellular matrix through these pathways. Explanation: NFs: normal blasts, CAFs: cancer−associated blasts, ECM: Extracellular matrix. “(+)”: promote, “(—)”: inhibit.

**Figure 3 cells-11-01780-f003:**
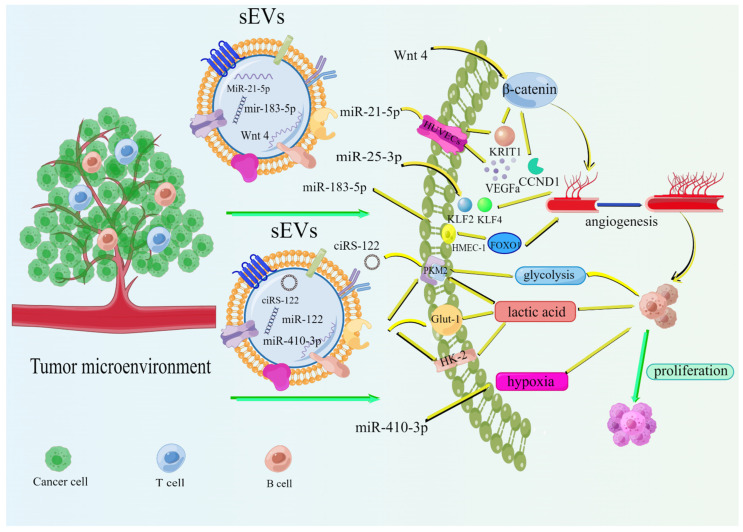
Mechanisms by which sEVs affect cell metabolism. When sEVs reach distant organs, Wnt 4 and three other miRNAs can promote angiogenesis by regulating proteins. Additionally, circ-IARS, the miRNA carried by CRC-derived sEVs, promotes the production of glucose lactate by upregulating glut-1, HK-2, and PKM-2 levels. In addition, miR-410-3p can promote the formation of an anoxic microenvironment. Ultimately, these cells metabolize and promote tumor proliferation. Explanation: VEGF: Vascular endothelial growth factor, HUVECs: human umbilical vein endothelial cells.

**Table 1 cells-11-01780-t001:** Mechanisms of different exosomes in the diagnosis of CRC.

Category	Name	Expression	Use in Colorectal Cancer	References
miRNA lncRNA	miR-92b	Downward adjustment	Distinguishing CRC from CA	[88]
	miR-125a-3p	Upward adjustment	Inhibition of migration and invasion	[86]
	miR-6803-5p miR-27a miR-130a	Upward adjustment Upward adjustment Upward adjustment	Suggests poor prognosis Differentiating CRC patients Differentiating CRC patients	[81] [83] [84]
	LOC344887 LINC00675 DPP10-AS HAGLR GAS5	Upward adjustment Upward adjustment Upward adjustment Upward adjustment Upward adjustment	Differentiating between CRC patients Differentiating between CRC patients Differentiating between CRC patients Differentiating between CRC patients Inhibits cell proliferation, migration, and invasion	[85] [85] [85] [85] [94]
	CCAT2	Upward adjustment	Enhanced invasion and metastasis	[95]
	CRNDE-h	Upward adjustment	Differentiating between patients with CRC and those with benign colorectal disease	[93]
Protein	LNCV6_116109	Upward adjustment	Differentiating between CRC patients	[92]
	LNCV6_98390	Upward adjustment	Differentiating between CRC patients	[92]
	LNCV6_84003	Upward adjustment	Differentiating between CRC patients	[92]
	CD147 Hsp60	Upward adjustment Upward adjustment	Differentiating between CRC patients Induced anti-tumor response	[91] [99]
	GPC1	Upward adjustment	Differentiating between CRC patients	[100]
	CPNE3	Upward adjustment	Differentiating colorectal cancer from other tumors	[101]
	SPARC	Upward adjustment	Promotes invasion and induces angiogenesis	[102]
	CEA	Upward adjustment	Monitoring for CRC recurrence	[103]

**Table 2 cells-11-01780-t002:** Clinical trials related to the use of sEVs for the diagnosis and treatment of CRC.

NCT Number	Title	Status	Study Results	Interventions	Characteristics
NCT04523389	Contents of Circulating Extracellular Vesicles: Biomarkers in Colorectal Cancer Patients	Recruiting	No Results Available	Biological: analysis (protein, lipid, RNA…) of circulating exosomes, size, and numberOther: Gathering additional information about the patient’s cancerDiagnostic Test: Diagnostic test	
NCT04298398	Impact of Group Psychological Interventions on Extracellular Vesicles in People Who Had Cancer	Not yet recruiting	No Results Available	Behavioral: Mindfulness Based-Cognitive Therapy (MBCT)Behavioral: Emotion-Focused Therapy Group for Cancer Recovery (EFT-CR)Other: Treatment as usual (no intervention)	Phase:Not Applicable
NCT04394572	Identification of New Diagnostic Protein Markers for Colorectal Cancer	Recruiting	No Results Available	Biological: Blood sample	
NCT01294072	Study Investigating the Ability of Plant Exosomes to Deliver Curcumin to Normal and Colon Cancer Tissue	Recruiting	No Results Available	Dietary Supplement: curcuminDietary Supplement: Curcumin conjugated with plant exosomesOther: No intervention	Phase:Phase 1
NCT04852653	A Prospective Feasibility Study Evaluating Extracellular Vesicles Obtained by Liquid Biopsy for Neoadjuvant Treatment Response Assessment in Rectal Cancer	Not yet recruiting	No Results Available	Procedure: Supplementary blood samples collection during the normal follow up of the patients

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
