# Peer review of "Small Extracellular Vesicles: Key Forces Mediating the Development and Metastasis of Colorectal Cancer"

_cells, 2022, doi:10.3390/cells11111780_

Round 1
Reviewer 1 Report
Minor comments:
- Please make sure English uppercase and lowercase: a. “Small” extracellular vesicles in Abstract; b. “ Pd-l1”, “Conde-h” (but “CRNDE-h” in Figure 1), “stat3”, and “g-MDSC”, “Mir” in Figure 1 legend; c. “Ccnd1” in line 184 (section 2.3); “Can” in line 276 (section 4).
- Please move the abbreviation explanation to the end of the figure legend.
- Please check your table 1 title: “can”.
- The gene names are blurrily shown in the sEV circle in Figure 2.
- You mention miR-27a and miR-130a in line 291 (section 5.1) but why they are not included in Table 1. In contrast, miR-125a-3p and miR-68.3-5p shown in Table 1 are not described in the article. The same problems appear for IncRNAs and Proteins in Table 1.
Author Response
Part 1 (Reviewer 1):
Dear Reviewers: Thank you very much for your reply and help. Thanks a lot for the reviewers’ comments and their kind suggestions of our manuscript (cells-1727520) entitled “Small extracellular vesicles: Key forces mediating the development and metastasis of colorectal cancer”. We provide this rebuttal letter to explain, point by point, the details of our revisions in the manuscript and our responses to the reviewers’ comments as follows. In order to make the changes easily viewable for you and the reviewers, in the revised paper, we marked the revision with red color. Besides, we have carefully checked through the whole manuscript and corrected some grammar mistakes. We are deeply grateful to you for your eager care and help. We hope the revised paper would satisfy you and the reviewers.Due to formatting issues with the figures and tables, we have uploaded our updated figures and tables in the attachment.
Minor comments
1.Q: Please make sure English uppercase and lowercase: a. “Small” extracellular vesicles in Abstract; b. “ Pd-l1”, “Conde-h” (but “CRNDE-h” in Figure 1), “stat3”, and “g-MDSC”, “Mir” in Figure 1 legend; c. “Ccnd1” in line 184 (section 2.3); “Can” in line 276 (section 4).
A: Thank you for taking the time to review our paper. Thanks to your suggestions, we re-examined our manuscript carefully and corrected the English capitalization and lowercase errors that appeared in the manuscript.
a. “Small” extracellular vesicles-> “small extracellular vesicle” (On page 1). b. “Pd-l1” -> “PD-L1”, “Conde-h” -> “CRNDE-h”, “stat3”-> “STAT3”, “g-MDSC”-> “G-MDSC”, “Mir”-> “mir”. (On page 4) c. “cnd1”->CCND1 (On page 6); “Can” -> “can” (On page 9).
2.Q: Please move the abbreviation explanation to the end of the figure legend.
A: Thank you. We have checked the full text and corrected our errors. We have moved the abbreviation explanation to the end of the figure legend. They are also marked in red. (On page 4,5,7).
3. Q: Please check your table 1 title: “can”.
A: Thank you. We read the relevant sentences carefully and corrected the error. “can” should be “CRC”. This sentence should be “Mechanisms of different exosomes in the diagnosis of CRC”. It has been marked in red in the text. (On page 9).
4.Q: The gene names are blurrily shown in the sEV circle in Figure 2.
A: Thank you. We double-checked our drawing of Figure 2 and found that it did have problems with clarity. We have redrawn Figure 2 and attached it below.
5.Q: You mention miR-27a and miR-130a in line 291 (section 5.1) but why they are not included in Table 1. In contrast, miR-125a-3p and miR-68.3-5p shown in Table 1 are not described in the article. The same problems appear for IncRNAs and Proteins in Table 1.
A: Thank you. We have carefully reviewed this section and corrected inconsistencies in the text and in Table 1 regarding miRNA, lncRNA, and protein. It has been marked in red in the text. (On page 9).
5.1. sEVs as early markers for colorectal cancer diagnosis
sEVs contain unique molecules, including DNA, RNA, microRNA and lncRNA, proteins, and lipids. As a result, they can be used as biomarkers for cancer diagnosis and to show specific gene expression profiles [85]. Biomarkers may be molecules secreted by tumors or specific responses of the body due to the presence of cancer. There are widespread plasma and other components in the body, making sEVs mobile and thus can play a crucial role in identifying and predicting biomarkers of human cancers. As a highly enriched miRNA in colorectal cancer sEVs, high expression of plasma exosomal miR-125a-3p and miR-6803-5p can increase the indication power and act a biomarker for the analysis in the initial CRC stages [86 87]. Plasma sEV miR-92b is barely expressed in CRC patients and highly expressed in healthy individuals and can be used as a biomarker for the diagnosis of CRC [88]. Plasma sEV miR-27a and miR-130a can be used as non-invasive biomarkers for early detection and prognosis prediction of CRC [89 90]. A recent raw letter study conducted, by screening key pathways and GO terms regarding up-and down-regulated transcripts, identified diagnostic CRC biomarkers lncRNA BLACAT1 and four down-regulated lncRNAs (LOC344887, LINC00675, DPP10-AS, and HAGLR) [91]. Another study showed that three exosomal lncRNA in plasma (LNCV6_116109, LNCV6_98390, LNCV6_84003) are expressed completely differently in CRC patients and healthy individuals and can be used as diagnostic biomarkers [92]. Liu et al. showed that exosomal lncRNA CRNDE-h expression is higher in the serum of patients with colonic adenomas than in normal subjects and thus could be used as a biomarker for the CRC diagnosis [93]. sEV carries the highly expressed lncRNA CCAT2 that promotes CRC metastasis and progression, while the lncRNA GAS5, whose high expression will inhibit CRC tumor cell proliferation and invasion. Both are expressed dramatically differently in CRC patients and healthy individuals and can be used as diagnostic and prognostic biomarkers for CRC [94 95]. Furthermore, Paul et al. demonstrated that the surface density of hyaluronic acid (HA) is significantly higher in sEVs from colorectal cancer cells than in normal cells using high-resolution atomic force microscopy (AFM) and spectroscopy (AFS) techniques [96]. This suggests that sEVs from colorectal cancer cells are different from colonic epithelial cells at the level of single vesicles and thus could be used as a potential biomarker for CRC diagnosis. Tian et al. also analyzed blood samples using HSFCM found that the level of CD147 positivity in sEVs was significantly higher in colorectal cancer patients than in healthy controls [97]. As a platform for the analysis and quantification of surface proteins in individual EVs, HSFCM has greatly enhanced the understanding of exosome-mediated intercellular communication and the development of advanced diagnostic strategies. A study showed that 36 proteins are upregulated and involved in the regulation of the metastatic microenvironment, while 22 proteins are downregulated and involved in tumor cell production and development in serum sEVs of CRC patients [98]. Another study indicated that heat shock protein 60 (Hsp60) from plasma sEVs can be used for CRC diagnosis [99]. The sEV-derived protein GPC1 is abundantly expressed in CRC patients with high sensitivity, while sEV CPNE3 is abundantly expressed in CRC tissues and less expressed in other cancers with relatively high specificity. Based on the biological properties of both, they can be used as biomarkers for the diagnosis of CRC [100 101]. sEV carries the protein SPARC, and its high expression often suggests a poor prognosis for CRC [102]. Plasma sEV-derived CEA has high sensitivity and specificity in CRC diagnosis and can be used as an early biomarker for the diagnosis of CRC [103]. Herein, we summarize the value of different sEVs in the diagnosis of colorectal cancer. (Table.1).

Reviewer 2 Report
This is a literature review concerning colorectal cancer
Apart from the limitation due to the nature and design of the study I have the following comments:
- Authors should highlight the clinical implications of their article (by adding a specific section)
- References must be updated improving the definition of tumor microenvironment: - Therapeutic Targets and Tumor Microenvironment in Colorectal Cancer. J Clin Med. 2021 May 25;10(11):2295. doi: 10.3390/jcm10112295; - The Role of Tumor Microenvironment Cells in Colorectal Cancer (CRC) Cachexia. Int J Mol Sci. 2021 Feb 4;22(4):1565. doi: 10.3390/ijms22041565
- The quality of the figures needs to be improved
Author Response
Major revisions
Dear Reviewers: Thank you very much for your reply and help. Thanks a lot for the reviewers’ comments and their kind suggestions of our manuscript (cells-1727520) entitled “Small extracellular vesicles: Key forces mediating the development and metastasis of colorectal cancer”. We provide this rebuttal letter to explain, point by point, the details of our revisions in the manuscript and our responses to the reviewers’ comments as follows. In order to make the changes easily viewable for you and the reviewers, in the revised paper, we marked the revision with red color. Besides, we have carefully checked through the whole manuscript and corrected some grammar mistakes. We are deeply grateful to you for your eager care and help. We hope the revised paper would satisfy you and the reviewers.Due to formatting issues with the figures and tables, we have uploaded our updated figures and tables in the attachment.
Part 2 (Reviewer 2): This is a literature review concerning colorectal cancer. Apart from the limitation due to the nature and design of the study I have the following comments:
1.Q: Authors should highlight the clinical implications of their article (by adding a specific section)
A: Thank you for taking the time to review our paper. We have carefully re-examined our manuscript and reviewed a large amount of relevant literature. We added a discussion of the clinical implications of our paper. The new content is listed below and it has been highlighted in red.
5.3 Clinical implications of sEV as a marker and drug carrier in CRC screening, diagnosis, treatment, and prognosis.
sEV has very obvious advantages as a biomarker for early diagnosis, diagnostic staging, efficacy monitoring, prognostic assessment of CRC. 1) 1) sEV has a very high clinical value in the early screening of CRC. Ren et al. found that: sEV-derived miR-196b-5p was highly differentially expressed in healthy individuals and CRC patients, where sEV-derived miR-196b -5p was highly expressed in CRC patients, but hardly expressed in healthy individuals, and its accuracy of screening for CRC was 0.88[117]. 2) sEV showed advantages far beyond other biomarkers in the diagnosis of CRC. Donatella et al. found that sEV was more sensitive than circulating tumor DNA (ctDNA) and circulating tumor cells (CTC) in the diagnosis of CRC, with higher sensi-tivity and specificity [118]. Furthermore, human plasma sEV-derived miRNAs had higher specificity than total plasma miRNAs in the early diagnosis of colon cancer. Li et al. found that plasma sEV miRNA (Let-7b-3p, miR-130-3P, miR-145-3P, miR-139-3p) were much more accurate than total plasma miRNAs for the early diagnosis of CRC patients The area under the curve (AUC) for the four markers combined was 0.932[119]. Another study showed that the sensitivity of sEV-derived (miR-1246, miR-23a) in CRC diagnosis was 92% and 95.5%, respectively [120]. 3) sEV has great potential as a biomarker for CRC prognostic analysis. Tan et al. found that serum sEV miRNA-199a, whose reduced expression showed a positive correlation with poor CRC prognosis and patients tended to have lower survival rates [121]. Another similar study showed that the sEV-carrying lncRNA MALAT1, at high expression, showed a significant correlation with poor patient prognosis, and its high expression was strongly as-sociated with poor quality of life and resistance to chemotherapeutic agents (Oxaliplatin) in patients with advanced CRC [122]. In CRC patients, increased levels of sEV miR-429 expression were significantly and positively correlated with patients having lower OS and poor prognosis [89] .4) sEV can be used as a drug carrier and ribonucleic acid therapeutic agent for the treatment of CRC. sEV, as a special targeted drug delivery system, can facilitate more drug flow to tumor cells, avoiding phagocytosis by macrophages and cleavage by some special enzymes. In the treatment to prevent the development of CRC, the use of targeted delivery of sEV carrying Adriamycin to the target organ can have a significant preventive effect and the incidence of CRC is greatly reduced [104]. When loaded with paclitaxel on milk-derived sEVs, modified by relevant techniques, and then infused back into humans, the tumor suppressive ability of paclitaxel was significantly enhanced and produced fewer side effects [123]. sEVs delivering small molecule RNAs have shown great value in CRC therapy. sEVs rely on small molecule RNA therapies such as small interfering (siRNA) and miRNA therapies. sEVs are used as nano-delivery systems loaded with siRNAs for targeted delivery to target cells. The sEV-based delivery systems are far superior to other delivery systems in inhibiting tumor growth [124]. Loading miRNA-21 and chemotherapeutic drugs (5-fluorouracil (5-FU)) onto sEVs significantly enhanced the anti-tumor effect of 5-FU on CRC [125]. Through nanoengineering techniques, sEVs carrying miR-214 could significantly enhance the sensitivity of radiotherapy and inhibit the growth of CRC [106]. The sEVs loaded with miR-25-3p inhibitor could significantly inhibit the vascular permeability of CRC tumor tissues and suppress tumor growth and metastasis in CRC patients [55]. There are already a large number of clinical trials investigating the clinical role of sEV in the early detection, diagnosis, efficacy monitoring, and assessment of prognosis of CRC. For example, a search through Clinicaltrials.gov, NCT04523389, NCT04394572, NCT01294072 and other relevant clinical trials are focusing on the clinical feasibility of applying sEV for CRC diagnosis, prognosis and drug carriers for treatment. Of course, sEVs currently have some uncertain risk factors, such as whether they will continue to be efficacious and safe for long-term application. Purification and isolation of sEV is also a big challenge at present, and these limitations deserve to be explored in future ongoing research.
References
- Zeng Z, Li Y, Pan Y, et al. Cancer-derived exosomal miR-25-3p promotes pre-metastatic niche formation by inducing vascular permeability and angiogenesis. Nature communications 2018;9(1):5395 doi: 10.1038/s41467-018-07810-w[published Online First: Epub Date]|.
- Dong SJ, Cai XJ, Li SJ. The Clinical Significance of MiR-429 as a Predictive Biomarker in Colorectal Cancer Patients Receiving 5-Fluorouracil Treatment. Medical science monitor : international medical journal of experimental and clinical research 2016;22:3352-61 doi: 10.12659/msm.900674[published Online First: Epub Date]|.
- Ren D, Lin B, Zhang X, et al. Maintenance of cancer stemness by miR-196b-5p contributes to chemoresistance of colorectal cancer cells via activating STAT3 signaling pathway. Oncotarget 2017;8(30):49807-23 doi: 10.18632/oncotarget.17971[published Online First: Epub Date]|.
- Verbanac D, Čeri A. Profiling Colorectal Cancer in the Landscape Personalized Testing-Advantages of Liquid Biopsy. 2021;22(9) doi: 10.3390/ijms22094327[published Online First: Epub Date]|.
- Min L, Zhu S, Chen L, et al. Evaluation of circulating small extracellular vesicles derived miRNAs as biomarkers of early colon cancer: a comparison with plasma total miRNAs. Journal of extracellular vesicles 2019;8(1):1643670 doi: 10.1080/20013078.2019.1643670[published Online First: Epub Date]|.
- Ogata-Kawata H, Izumiya M, Kurioka D, et al. Circulating exosomal microRNAs as biomarkers of colon cancer. PloS one 2014;9(4):e92921 doi: 10.1371/journal.pone.0092921[published Online First: Epub Date]|.
- Tan HY, Zheng YB, Liu J. Serum miR-199a as a potential diagnostic biomarker for detection of colorectal cancer. European review for medical and pharmacological sciences 2018;22(24):8657-63 doi: 10.26355/eurrev_201812_16630[published Online First: Epub Date]|.
- Li P, Zhang X, Wang H, et al. MALAT1 Is Associated with Poor Response to Oxaliplatin-Based Chemotherapy in Colorectal Cancer Patients and Promotes Chemoresistance through EZH2. Molecular cancer therapeutics 2017;16(4):739-51 doi: 10.1158/1535-7163.mct-16-0591[published Online First: Epub Date]|.
- Agrawal AK, Aqil F, Jeyabalan J, et al. Milk-derived exosomes for oral delivery of paclitaxel. Nanomedicine : nanotechnology, biology, and medicine 2017;13(5):1627-36 doi: 10.1016/j.nano.2017.03.001[published Online First: Epub Date]|.
- Kamerkar S, LeBleu VS, Sugimoto H, et al. Exosomes facilitate therapeutic targeting of oncogenic KRAS in pancreatic cancer. Nature 2017;546(7659):498-503 doi: 10.1038/nature22341[published Online First: Epub Date]|.
- Liang G, Zhu Y, Ali DJ, et al. Engineered exosomes for targeted co-delivery of miR-21 inhibitor and chemotherapeutics to reverse drug resistance in colon cancer. Journal of nanobiotechnology 2020;18(1):10 doi: 10.1186/s12951-019-0563-2[published Online First: Epub Date]|.
2.Q: References must be updated improving the definition of tumor microenvironment: - Therapeutic Targets and Tumor Microenvironment in Colorectal Cancer. J Clin Med. 2021 May 25;10(11):2295. doi: 10.3390/jcm10112295; - The Role of Tumor Microenvironment Cells in Colorectal Cancer (CRC) Cachexia. Int J Mol Sci. 2021 Feb 4;22(4):1565. doi: 10.3390/ijms22041565
A: Thank you for your corrections on the shortcomings of our paper. We have re-examined our content on the definition of TME and found that there are indeed huge deficiencies. We have reviewed the relevant literature that you provided us with and have also reviewed a large amount of other literature. We have refined our definition of TME and the new content is as follows and it has been highlighted in red.
TME is a complex, dynamic network of cells consisting of tumor cells, various types of immune cells (T cells, B cells, tumor macrophages, NK cells), fibroblasts, adipocytes, extracellular matrix, microbiota, and a variety of cytokines and metabolites [4 5]. TME promotes the development and proliferation of tumor cells by recruiting immunosuppressive cells, remodeling the extracellular matrix and promoting angiogenesis [6]. Furthermore, abnormalities such as hypoxia, metabolic disturbances and oxidative stress are evident in TME. These abnormalities further suppress normal immune function and promote stromal fibrosis, which ultimately leads to tumor cell proliferation and metastasis. In CRC, there is little and phenotypically dysregulated infiltration of normal T and NK cells, resulting in suppression of normal anti-tumor functions [7]. At the same time, myeloid suppressor cells and tumor-associated neutrophils (TAN), which are recruited in large numbers in the TME, suppress normal immune function by activating metalloproteinases, creating an immunosuppressive microenvironment [8]. In addition, pro-angiogenic factors secreted by CRC tumor cells promote the generation of dense vascular tissue, which together with the dense fibrotic network leads to hypoxia and metabolic disturbances in TME [9]. These factors interact and ultimately promote CRC proliferation and metastasis.
References
- Gallo G, Vescio G, De Paola G. Therapeutic Targets and Tumor Microenvironment in Colorectal Cancer. 2021;10(11) doi: 10.3390/jcm10112295[published Online First: Epub Date]|.
- Hanus M, Parada-Venegas D, Landskron G, et al. Immune System, Microbiota, and Microbial Metabolites: The Unresolved Triad in Colorectal Cancer Microenvironment. Frontiers in immunology 2021;12:612826 doi: 10.3389/fimmu.2021.612826[published Online First: Epub Date]|.
- Kasprzak A. The Role of Tumor Microenvironment Cells in Colorectal Cancer (CRC) Cachexia. International journal of molecular sciences 2021;22(4) doi: 10.3390/ijms22041565[published Online First: Epub Date]|.
- Rocca YS, Roberti MP, Arriaga JM, et al. Altered phenotype in peripheral blood and tumor-associated NK cells from colorectal cancer patients. Innate immunity 2013;19(1):76-85 doi: 10.1177/1753425912453187[published Online First: Epub Date]|.
- Germann M, Zangger N, Sauvain MO, et al. Neutrophils suppress tumor-infiltrating T cells in colon cancer via matrix metalloproteinase-mediated activation of TGFβ. 2020;12(1):e10681 doi: 10.15252/emmm.201910681[published Online First: Epub Date]|.
- Albini A, Bruno A, Noonan DM, et al. Contribution to Tumor Angiogenesis From Innate Immune Cells Within the Tumor Microenvironment: Implications for Immunotherapy. Frontiers in immunology 2018;9:527 doi: 10.3389/fimmu.2018.00527[published Online First: Epub Date]|.
3.Q: The quality of the figures needs to be improved
A: Thank you. We carefully examined our related pictures and found that it does have problems. We have recreated the related image to make it look clearer and more beautiful. Thank you for your corrections and help with our manuscript.

Round 2
Reviewer 2 Report
I'm satisfied with the changes made
This manuscript is a resubmission of an earlier submission. The following is a list of the peer review reports and author responses from that submission.